# SegNeRF: 3D Part Segmentation with Neural Radiance Fields

## Abstract

Recent advances in Neural Radiance Fields (NeRF) boast impressive performances for generative tasks such as novel view synthesis and 3D reconstruction. Methods based on neural radiance fields are able to represent the 3D world implicitly by relying exclusively on posed images. Yet, they have seldom been explored in the realm of discriminative tasks such as 3D part segmentation. In this work, we attempt to bridge that gap by proposing SegNeRF: a neural field representation that integrates a semantic field along with the usual radiance field. SegNeRF[1] inherits from previous works the ability to perform novel view synthesis and 3D reconstruction, and enables 3D part segmentation from a few images. Our extensive experiments on PartNet show that SegNeRF is capable of simultaneously predicting geometry, appearance, and semantic information from posed images, even for unseen objects. The predicted semantic fields allow SegNeRF to achieve an average mIoU of **30.30%** for 2D novel view segmentation, and **37.46%** for 3D part segmentation, boasting competitive performance against point-based methods by using only a few posed images. Additionally, SegNeRF is able to generate an explicit 3D model from a single image of an object taken in the wild, with its corresponding part segmentation.

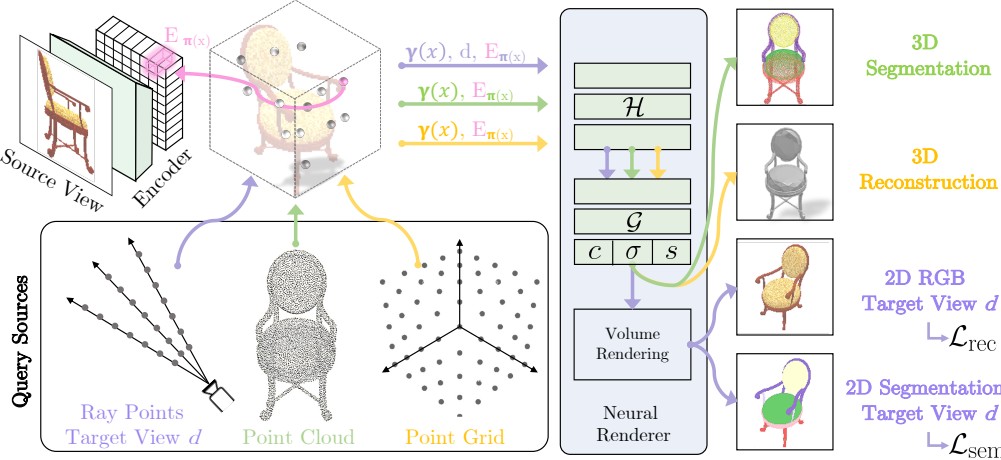

Figure 1: **SegNeRF framework:** Implicit Representation with Neural Radiance Fields for 2D novel view Semantic Segmentation, as well as 3D Segmentation and Reconstruction. Our model takes as input one or more source views of an object (top-left image). The source view is used to generate a feature grid, which is queried with a set of (i) ray points for volume rendering, (ii) an object point cloud for 3D semantic part segmentation, or (iii) a point grid for 3D reconstruction. Training is supervised only through images in the form of 2D reconstruction and segmentation losses. However, at test time, our model is also capable of generating 3D semantic segmentation and reconstruction.

---

[1] We will release our code publicly for reproducibility.

# 1 INTRODUCTION

Humans are able to *perceive* and *understand* the objects that surround them. Impressively, we understand objects even though our visual system only perceives partial information through 2D projections, *i.e.* images. Despite possible occlusions, a single image allows us to infer an object's geometry, estimate its pose, and recognize its parts and their location in a 3D space. Inspired by this capacity of using 2D projections to understand 3D objects, we aim to understand object part semantics in 3D space by solely relying on image-level supervision.

Most works learn object part segmentation by leveraging 3D supervision, *i.e.* point clouds or meshes (Wang & Lu, 2019; Qi et al., 2017b; Li et al., 2018). However, given the accessibility of camera sensors, it is easier at inference time to have access to a collection of images rather than a 3D scan of the object. To exploit real-world scenarios, it would thus be advantageous to have the ability of understanding 3D information from image-level supervision. For that purpose, Neural Radiance Fields (NeRF, Mildenhall et al., 2020) emerged as a cornerstone work in novel-view synthesis. NeRFs showed how to learn a 3D object's representation solely based on image supervision. While NeRF's seminal achievement was learning individual scenes with remarkable details, its impact grew beyond the single-scene setup, spurring follow-up works on speed (Müller et al., 2022; Yu et al., 2021a), scale (Martin-Brualla et al., 2021; Xiangli et al., 2021), and generalization (Jang & Agapito, 2021; Yu et al., 2021b). Despite the rapid advances in NeRF, only a handful of works (Zhi et al., 2021; Vora et al., 2021) have leveraged volume rendering for semantic segmentation or 3D reconstruction. Yet, these capabilities are a defining feature of 3D understanding, with potential applications in important downstream tasks.

In this work, we present SegNeRF: an implicit representation framework for simultaneously learning geometric, visual, and semantic fields from posed images, as shown in Figure 1. At training time, SegNeRF only requires paired posed colored images and semantic masks of objects, inheriting NeRF's independence of direct geometric supervision. At inference time, SegNeRF projects appearance and semantic labels on multiple views, and can thus accommodate a variable number of input views without expensive on-the-fly optimization. SegNeRF inherits the ability to perform 3D reconstruction and novel view synthesis from previous works (PixelNeRF, Yu et al., 2021b). However, we demonstrate the ability to learn 3D semantic information through volume rendering by leveraging a semantic field. We validate SegNeRF's performance on PartNet (Mo et al., 2019), a large-scale dataset of 3D objects annotated with semantic labels, and show that our approach performs on-par with point-based methods without the need for any 3D supervision.

**Contributions.** We summarize our contributions as follows. **(i)** We present SegNeRF, a versatile 3D implicit representation that jointly learns appearance, geometry, and semantics from posed RGB images. **(ii)** We provide extensive experiments validating the capacity of SegNeRF for 3D part segmentation despite relying exclusively on image supervision during training. **(iii)** To the best of our knowledge, SegNeRF is the first multi-purpose implicit representation capable of jointly reconstructing and segmenting novel objects without expensive test-time optimization.

# 2 RELATED WORK

## 2.1 3D GEOMETRICAL REPRESENTATIONS

The classical representations for data-driven 3D learning systems can be divided into three groups: voxel-, point-, and mesh-based representations. Voxels are a simple 3D extension of pixel representations; however, their memory footprint grows cubically with resolution (Brock et al., 2016; Gadelha et al., 2017). While point clouds are more memory-efficient, they need post-processing to recover missing connectivity information (Fan et al., 2017; Achlioptas et al., 2018). Most mesh-based representations do not require post-processing, but they are often based on deforming a fixed size template mesh, hindering the processing of arbitrary shapes (Kanazawa et al., 2018; Ranjan et al., 2018). To alleviate these problems, there has been a strong focus on implicitly representing 3D data via neural networks (Mescheder et al., 2019; Park et al., 2019; Sitzmann et al., 2019; Mildenhall et al., 2020). The fundamental idea behind these methods is using a neural network $f(\mathbf{x})$ to model certain physical properties (*e.g.*, occupancy , distance to the surface, color, density, or illumination) for all points $\mathbf{x}$ in 3D space.

Most recent efforts have developed differentiable rendering functions that allow neural implicit shape representations to be optimized using only 2D images, removing the need for ground truth 3D shapes (Liu et al., 2019b; Niemeyer et al., 2020b; Yariv et al., 2020; Sitzmann et al., 2019). For instance, Sitzmann et al. (2019) propose Scene Representation Network (SRN), a recurrent neural network that marches along each ray to decide where the surface is located while learning latent codes across scene instances. Neural Radiance Fields (NeRF, Mildenhall et al., 2020) encodes a scene with 5D radiance fields, which is accurately rendered to high-resolution RGB information per light ray. PixelNeRF (Yu et al., 2021b) extended NeRF to work on novel scenes and object categories by using an image encoder to condition a NeRF on image features to dispose of test-time optimization. In this paper, we treat PixelNeRF as a powerful generalizable 3D implicit representation, and extend it to learn a semantic field along with the usual radiance field. This extension allows us to (1) predict 2D novel view semantics through volume rendering, (2) efficiently query the density and semantic fields with a bounded 3D volume or arbitrary 3D points to obtain both 3D object reconstruction and part segmentation.

## 2.2 3D SEMANTIC SEGMENTATION

We discuss previous methods segmenting 3D data and focus on those aimed at 3D part segmentation.

**Explicit Representations.** Most existing works for 3D semantic segmentation are point-, voxel- and multi-view-based approaches. As for point-based methods, the seminal PointNet (Qi et al., 2017a) work excels at extracting global information, but its architecture limits its ability to encode local structures. Various studies addressed this issue (Qi et al., 2017b; Li et al., 2018), proposing different aggregation strategies of local features. Voxel-based methods achieve competitive results (Le & Duan, 2018; Song et al., 2017). However, they require high-resolution voxels with more detailed structure information for fine-grained tasks like part segmentation, which leads to high computation costs. In contrast to these methods, we only use 2D supervision and inputs to segment a dense 3D representation. Kalogerakis et al. (2017) acquire a set of images from several viewpoints that best cover the object surface and then predict and revise part labels independently using multi-view Fully Convolutional Networks and surface-based Conditional Random Fields. Nevertheless, it requires a ground truth 3D object to query points from at test time, while our method can reconstruct the object while segmenting it.

**Implicit Representations.** Similar to our approach, Semantic NeRF (Zhi et al., 2021), NeSF (Vora et al., 2021), and Semantic SRN (Kohli et al., 2020) leverage implicit representations to predict semantic labels. Both NeRF-based approaches (*i.e.* Semantic NeRF and NeSF) succeed at this task by predicting, in addition to per-point radiance and density, the point's semantic class. However, both approaches are limited. Specifically, Semantic NeRF does not generalize to novel scenes. Moreover, NeSF neglects color information, is not validated in a standard dataset, and requires test-time optimization to generalize to novel scenes. On the other hand, Semantic SRN conducts unsupervised feature learning from posed RGB images and then learns to output semantic segmentation maps. Compared to these works, our method generalizes to novel views and objects, does not rely on per-scene test-time optimization, requires only relative camera poses, and is validated on a well-established benchmark (*i.e.* PartNet).

## 2.3 3D RECONSTRUCTION FROM IMAGES

**Multiple-view reconstruction.** Reconstructing a 3D scene from a limited set of 2D images is a long-standing challenge in computer vision (Hartley & Zisserman, 2003). Modern learning-based methods (Hartmann et al., 2017; Donne & Geiger, 2019; Huang et al., 2018) can reconstruct with just a few views by leveraging shape priors learnt from large datasets. However, to perform such reconstruction, these methods need to train on costly and explicit 3D supervision. In contrast, SegNeRF can provide dense high-quality 3D geometry while only requiring 2D supervision.

**Single-view reconstruction.** Most single-view 3D reconstruction methods condition neural 3D representations on images (Niemeyer et al., 2020a; Xu et al., 2019; Liu et al., 2019a). For instance, Niemeyer et al. (2020a) uses an input image to condition an implicit surface-and-texture network. Liu et al. (2019a) learn to deform an initial mesh into a target 3D shape. Our SegNeRF method

shares this ability of reconstructing from a single view, however, it can also leverage a variable number of views to improve the reconstruction.

## 3 METHODOLOGY

We now present SegNeRF, a multi-purpose method for jointly learning consistent 3D geometry, appearance, and semantics. During training, we require multiple posed views of each object along with their corresponding semantic maps. While SegNeRF trains solely with image-level supervision, at inference time it can receive a single image of an unseen object and produce (i) 2D novel-view part segmentation, (ii) 3D part segmentation, and (iii) 3D reconstruction. Furthermore, SegNeRF can generate more accurate predictions by leveraging additional input posed images of the object.

Before detailing our method, SegNeRF, we first give an overview of volume rendering which allows us to learn in 3D space from images.

### 3.1 OVERVIEW OF NEURAL VOLUME RENDERING

Neural volume rendering relies on learning two functions: $\sigma(\mathbf{x}; \theta) : \mathbb{R}^3 \mapsto \mathbb{R}$ which maps a point in space $\mathbf{x}$ onto a density $\sigma$, and $c(\mathbf{x}, \mathbf{d}; \theta) : \mathbb{R}^3 \times \mathbb{R}^3 \mapsto \mathbb{R}^3$ that maps a point in space viewed from direction $\mathbf{d}$ onto a radiance $c$. The parameters $\theta$ that define the density and radiance functions are typically optimized to represent a single scene by using multiple posed views of the scene. In order to learn these functions, they are evaluated at multiple points along a ray $\mathbf{r}(t) = \mathbf{o} + t\mathbf{d}$, $t \in [t_n, t_f]$, defined by the camera origin $\mathbf{o} \in \mathbb{R}^3$, pixel viewing direction $\mathbf{d}$, and camera near and far clipping planes $t_n$ and $t_f$. A pixel color for the ray can then be obtained through volume rendering via:

$$\hat{C}(\mathbf{r}; \theta) = \int_{t_n}^{t_f} T(t)\, \sigma(\mathbf{r}(t))\, c(\mathbf{r}(t), \mathbf{d})\, \mathrm{d}t, \quad \text{where } T(t) = \exp\left(-\int_{t_n}^{t} \sigma(\mathbf{r}(s))\, \mathrm{d}s\right), \quad (1)$$

which can be computed with the original posed images. In practice, a summation at discrete samples along the ray is used instead of the continuous integral.

This volume rendering process allows us to supervise the learning of implicit functions $c$ and $\sigma$, which span 3D space by using only rendered images through the reconstruction loss:

$$\mathcal{L}_{\text{rec}}(\theta) = \frac{1}{|R|} \sum_{r \in R} \left\| C(r) - \hat{C}(r; \theta) \right\|_2^2, \quad (2)$$

where $R$ is the batch of rays generated from a random subset of pixels from training images.

### 3.2 PREDICTING VOLUME RENDERING

In the previous section, minimizing the reconstruction loss in equation 2 yields an implicit representation for a single scene. It is possible to obtain a model that generalizes to novel scenes by conditioning the learnt density and appearance on local feature vectors $f(I_i, \mathbf{x}, \mathbf{d})$ derived from each of the input source images $I_i$. In particular, given a set of images $\mathbf{I}$, it is possible to predict the radiance and density functions as:

$$c(\mathbf{I}, \mathbf{x}, \mathbf{d}) = \mathcal{G}_c\left(\frac{1}{|I|} \sum_{I_i \in \mathbf{I}} f(I_i, \mathbf{x}, \mathbf{d})\right), \quad \sigma(\mathbf{I}, \mathbf{x}, \mathbf{d}) = \mathcal{G}_\sigma\left(\frac{1}{|I|} \sum_{I_i \in \mathbf{I}} f(I_i, \mathbf{x}, \mathbf{d})\right), \quad (3)$$

where $\mathcal{G}_c$ and $\mathcal{G}_\sigma$ are MLPs that predict radiance and density fields for an object with images $\mathbf{I}$ at a given location and direction. The same reconstruction loss in equation 2 is minimized to obtain network parameters for both the feature extractor $f$ and the MLPs $\mathcal{G}_c$ and $\mathcal{G}_\sigma$.

Next, we elaborate on the inner workings of the feature extractor $f$. Each input image $I_i$ is first encoded into a feature map $E(I_i)$. Then, each query point $\mathbf{x}$ is projected onto every feature image, obtaining image coordinates $\pi_i(\mathbf{x})$ and leading to per-view feature vectors $E(I_i)_{\pi_i(\mathbf{x})}$. Query points are also projected onto $I_i$'s local coordinates via its associated projection matrix $P^{(i)}$. A positional

encoding is then applied onto the local coordinates, resulting in $\gamma(P^{(i)}\mathbf{x})$. The per-view feature vectors $E(I_i)_{\pi_i(\mathbf{x})}$ are concatenated with the positional-encoded local coordinates $\gamma(P^{(i)}\mathbf{x})$ and the viewing directions $\mathbf{d}$. The concatenated vectors are then processed by an MLP $\mathcal{H}$ to obtain per-view feature vectors $f(I_i, \mathbf{x}, \mathbf{d})$ as follows:

$$f(I_i, \mathbf{x}, \mathbf{d}) = \mathcal{H}(\mathrm{cat}[E(I_i)_{\pi_i(\mathbf{x})}, \gamma(P^{(i)}\mathbf{x}), \mathbf{d}]). \tag{4}$$

In summary, feature vectors across all input views are aggregated to obtain a single per-point feature vector $f(\mathbf{I}, \mathbf{x}, \mathbf{d})$, which is fed to the global MLP layer $\mathcal{G}$ to predict appearance and geometry as shown in equation 3.

### 3.3 SEMANTIC VOLUME RENDERING

We now describe our proposed extension to this framework, which enables our network to learn both 2D and 3D semantic segmentation from object images. In a similar fashion to how a radiance field represents radiance at each point in space, we can imagine a similar semantic field which represents the likelihood of an object part occupying each point in space. We propose to learn such a semantic field function $s(\mathbf{I}, \mathbf{x}, \mathbf{d}) \in \mathbb{R}^c$, where $c$ is the number of object part classes, using an MLP $\mathcal{G}_s$ similar to equation 3. In particular, function $s$ predicts a probability score for each location $\mathbf{x}$ and direction $\mathbf{d}$ conditioned on a set of images $\mathbf{I}$. Similar to the color rendering, we aggregate the semantic predictions along a ray for a given set of images by using the learnt density function as follows:

$$\hat{S}(\mathbf{I}, \mathbf{r}; \theta) = \int_{t_n}^{t_f} T(t)\, \sigma(\mathbf{r}(t))\, s(\mathbf{I}, \mathbf{r}(t), \mathbf{d})\mathrm{d}t. \tag{5}$$

A softmax function is used to convert the aggregated semantic scores $\hat{S}$ to pixel semantic probabilities. At last, the predicted pixel semantic probabilities are optimized using a cross entropy loss in the following form:

$$\mathcal{L}_{\mathrm{sem}}(\theta) = \frac{1}{|R|}\sum_{r \in R} \mathrm{CE}\left(S(r), \hat{S}(r; \theta)\right), \tag{6}$$

where $S(r)$ is the ground truth segmentation label for the pixel corresponding to ray $r$. We note that we train SegNeRF jointly for segmentation and image reconstruction through the combined loss

$$\mathcal{L}_{\mathrm{tot}}(\theta) = \mathcal{L}_{\mathrm{rec}}(\theta) + \lambda\mathcal{L}_{\mathrm{sem}}(\theta). \tag{7}$$

We emphasize that, upon convergence, a single forward pass through SegNeRF obtains the semantic field function $s$ which can be leveraged for point cloud segmentation applications. Experimentally, we show that the semantic field $s$, learnt using only image supervision, contains accurate 3D segmentation.

### 3.4 3D SEMANTIC SEGMENTATION

Next, we show how to leverage the learnt semantic field function $s$ for 3D semantic segmentation. The task of 3D semantic segmentation requires classifying every point on an object's surface with a single class $c$ out of a known set of classes $\mathbf{C}$. In practice, methods typically aim to classify a discrete set of points $\mathbf{X}$ sampled from the object's surface. This classification can be achieved by querying the semantic field function $s(\mathbf{I}, \mathbf{x}, \mathbf{d})$ at every point $\mathbf{x}$ in $\mathbf{X}$. The class for point $\mathbf{x}$ for a novel object captured by a set of images $\mathbf{I}$ is given by:

$$c(\mathbf{x}) = \arg\max s(\mathbf{I}, \mathbf{x}, \mathbf{0}). \tag{8}$$

Even though the semantic field function $s$ has been trained through volume rendering with image supervision, we show experimentally that the learnt field is consistent with the underlying 3D representation and thus can be used in a point-wise fashion for predicting 3D semantic segmentation.

Computing 3D semantic segmentation with this procedure is vastly more efficient than performing volume rendering, since classifying each point requires a single forward pass instead of aggregating multiple computations per ray as done for volume rendering. While the learnt semantic field $s$ represents semantic segmentation at every location in space, we extract segmentation at the same set of points $\mathbf{X}$ as point cloud segmentation methods for comparison purposes. However, this set of points is generally not required for computing the segmentation of a novel object. In practice, it is possible to extract semantic segmentation for an object without having access to a set of points by first extracting an object mesh, before querying the vertices of the extracted mesh for semantic segmentation. We demonstrate this capability by obtaining a semantic reconstruction of real objects in Section 5 from a single image without access to any 3D structure.

## 4 EXPERIMENTS

We experiment with our novel SegNeRF framework, showing its capability to generate novel 2D views with segmentation masks, reconstruct 3D geometry and segment 3D semantic parts of objects from one or more images.

**Datasets.** For all experiments, we utilize a subset of the PartNet (Mo et al., 2019) dataset that overlaps with the ShapeNet (Chang et al., 2015) dataset. PartNet contains an extensive collection of object meshes that have been divided into parts, along with their corresponding human-labeled part semantic segmentation at multiple resolutions. ShapeNet includes the corresponding colored 3D meshes, which we require to obtain the input renderings for our models. From each matched PartNet and ShapeNet object model, we first align and normalize the part and object meshes. We then render views from the colored mesh along with part segmentation masks rendered from the aligned part meshes using the PyTorch3D (Ravi et al., 2020) internal rasterizer. For the training set, we render 250 camera perspectives sampled uniformly at random on a unit sphere, with the object at the origin. To generate the validation and test sets, we render 250 camera perspectives sampled from an archimedean spiral covering the top half of each object instance, following previous work (Kohli et al., 2020). Due to the need for having both semantic parts and colored meshes, the number of annotated object instances is lower than in the original PartNet dataset keeping 81.2% of the shapes on average. We report experiments for objects with the highest number of instances and re-run the baselines on this PartNet subset for a fair comparison. We will release the RGB and semantic renderings publicly for reproducibility.

**Training Details.** Following PixelNeRF (Yu et al., 2021b), we train our proposed SegNeRF models using a batch size of 4 (objects) and 1024 sampled rays per object. We train all of our models on 4 Nvidia V100 GPUs for 600k steps, with a single model trained for each object category. We use a segmentation loss weight of $\lambda = 1$.

### 4.1 2D NOVEL-VIEW PART SEGMENTATION

Given a single or multiple posed views of an object unseen during training, we aim to obtain the segmentation mask for the given object at novel views. This task requires not only understanding the semantics of an object class in 2D space but also in 3D space to ensure consistent novel-view segmentation.

**Metrics.** We evaluate novel-view segmentation by using mean intersection over union (mIoU) over every part category for each object class. For single-view experiments, a front-facing image of the object (view ID 135) is used as input. A total of 25 novel views around each object are predicted and averaged for each model. These 25 views are equally spaced around the the archimedean spiral.

**Baselines.** For the PartNet 2D semantic segmentation task, we compare quantitatively and qualitatively with three baselines. First, we establish an upper bound by training a DeepLabv3 model in an unrealistic setting in which the model is granted access to all original observations, and performs segmentation on the test images without novel view prediction. Second, Semantic-SRN (Kohli et al., 2020), a method proposed by Kohli *et al.* which is the current state-of-the-art in few-shot novel-view semantic segmentation, but requires test-time optimization. Lastly, we proposed "PixelLab", a two-stage baseline composed of PixelNeRF (Yu et al., 2021b) for novel view synthesis and DeepLabv3 (Chen et al., 2017) for semantic segmentation. We use DeepLabv3, a popular

| | Avg | Bed | Bott | Chair | Clock | Dish | Disp | Ear | Fauc | Knife | Micro | Stora | Table | Trash | Vase |
|---|---|---|---|---|---|---|---|---|---|---|---|---|---|---|---|
| DeepLabv3 | 31.59 | 18.68 | 34.14 | 37.01 | 22.1 | 19.65 | 53.11 | 23.88 | 41.1 | 30.62 | 24.36 | 38.42 | 22.8 | 33.16 | 43.18 |
| PixelLab | 21.43 | 13 | 27.95 | 12.79 | 18.17 | 16.66 | **45.80** | **18.37** | 28.25 | 26.84 | 19.35 | 15.48 | 10.2 | 20.79 | 26.4 |
| Semantic SRN | - | - | - | 12.81 | - | - | - | - | - | - | - | - | 8.09 | - | - |
| SegNerf (Ours) | 30.30 | 19.65 | 35.06 | 26.63 | 25.66 | 34.14 | 45.57 | 17.13 | 35.68 | 31.26 | 33.07 | 32.03 | 14.17 | 31.18 | 42.91 |

Table 1: **Fine-grained 2D semantic segmentation for novel views (2D part-category mIoU%).** Quantitative results for fine-grained 3D semantic segmentation are presented, with a detailed breakdown by category. Our method outperforms the baseline methods by significant margins in most categories. Upper bound method (DeepLabv3) is highlighted in gray.

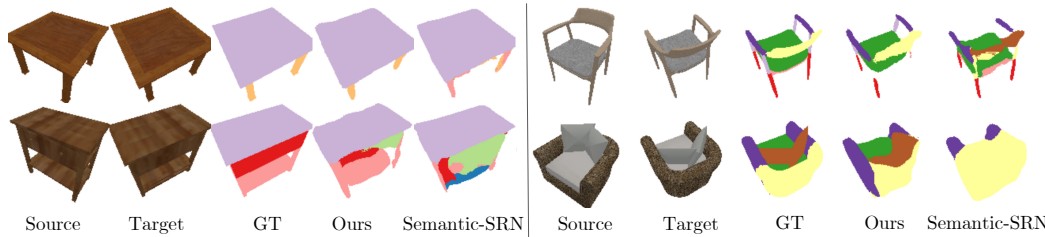

| Source | Target | GT | Ours | Semantic-SRN | Source | Target | GT | Ours | Semantic-SRN |

Figure 2: **2D semantic part segmentation for novel views.** Qualitative results for 2D semantic part segmentation for novel views.

image semantic segmentation baseline, with a ResNet50 backbone pre-trained on ImageNet and finetuned on our PartNet subset for 20 epochs using ADAM and a base learning rate of 1e-3. Once DeepLab v3 is trained, we evaluate it on the validation set generated by PixelNeRF consisting of 25 novel poses, and rendered from a single view of the object instance using a category-specific PixelNeRF. We refer the reader to the Appendix for further details.

**Results.** Quantitative. In Table 1, we show the results of 2D semantic segmentation for novel views. Our method exhibits better performance than Semantic-SRN and "PixelLab" for all categories. Lastly, as expected, the DeepLabv3 upper bound performs slightly better than SegNeRF (mIoU of 31.59% *vs*. 30.30%). The results of the coarse-grained 2D semantic segmentation for novel views are described in the Appendix. Qualitative. Semantic-SRN produces undesirable results, as shown in Figure 2. We attribute this phenomenon to Semantic-SRN's optimization relying only on RGB information, which does not match the model's training on RGB and semantic maps, introducing noise in the objects' latent code. Furthermore, Figure 2 also illustrates how PixelLab mislabels fine structures. Unlike PixelNeRF, SegNeRF does exploit the 3D representation while training, making it more apt for directly predicting semantic labels.

## 4.2 SINGLE/MULTI-VIEW 3D PART SEGMENTATION

The task of 3D part segmentation aims to assign a part category to each point in a given 3D point cloud representing a single object. Here, we aim to perform 3D part segmentation of a given point cloud by using *only* the information from specific posed images. The point cloud is only used as a query for segmentation to compare against point-based methods.

**Metrics.** We evaluate 3D part segmentation by calculating the mIoU over part categories for every object class. For single-view experiments, a front-facing image of the object (view ID 135) is used as input. Experiments using two images take as additional back-facing image (view ID 192), while experiments with four images include a top and a front view (view IDs 0, 249). Additional metrics such as accuracy and mean accuracy are reported in the Appendix.

**Baselines.** We compare our method with a fair multi-view baseline and more widely adopted 3D point-based methods. For the point-based methods, we train PointNet (Qi et al., 2017a), Point-Net++ (Qi et al., 2017b), and PosPool (Liu et al., 2020) on our dataset. Additionally, we propose a multi-view (MV) baseline in which a query point cloud is projected into DeepLabv3 predicted semantic segmentation masks. Every point is projected onto each view to obtain a per-view segmentation label. It is important to mention that the final label is computed by a voting method among

| | Avg | Bed | Bott | Chair | Clock | Dish | Disp | Ear | Fauc | Knife | Micro | Stora | Table | Trash | Vase |
|---|---|---|---|---|---|---|---|---|---|---|---|---|---|---|---|
| **PointNet** | 31.52 | 24.2 | 20.56 | 33.26 | 25.27 | 34.06 | 65.66 | 23.06 | 38.12 | 18.71 | 29.59 | 29.59 | 29.54 | 30.61 | 39.1 |
| **PointNet++** | 34.88 | 29.16 | 21.78 | 39.71 | 25.97 | 41.43 | 62.4 | 19.37 | 42.94 | 27.91 | 29.53 | 35.78 | 34.75 | 33.89 | 43.65 |
| **PosPool** | 46.42 | 43.2 | 25.5 | 48.46 | 41.44 | 62.86 | 66.53 | 27.18 | 50.5 | 31.19 | 54.99 | 47.04 | 43.99 | 52.11 | 54.95 |
| **MV (1)** | 22 | 12.08 | 36.01 | 24.41 | 18.38 | 14.96 | 36.41 | 15.31 | 34.12 | 25.43 | 17.29 | 13.46 | 11.45 | 18.96 | 29.7 |
| **MV (2)** | 25.16 | 12.96 | 37.61 | 31.09 | 19.29 | 13.78 | 51.39 | 17.21 | 37.65 | 27.13 | 16.04 | 16.6 | 16.03 | 22.33 | 33.16 |
| **MV (4)** | 25.23 | 15.38 | 30.82 | 34.2 | 16.71 | 11.47 | 53.13 | 18.04 | 37.65 | 30.63 | 14.76 | 15.96 | 17.09 | 22.94 | 34.44 |
| **MV (25)** | 27.43 | 18.21 | 37.91 | 36.26 | 19.12 | 15.1 | 53.37 | 24.89 | 40.74 | 27.46 | 15.35 | 21.3 | 17.29 | 22.16 | 34.79 |
| **SegNeRF (1)** | 32.44 | 15.99 | 41.05 | 34.83 | 24.09 | 39.73 | 57.43 | 21.36 | 44.25 | 41.08 | **24.79** | 22.71 | 15.58 | 30.44 | 40.87 |
| **SegNeRF (2)** | 36.09 | 18.93 | 42.67 | 37.93 | 26.45 | 44.98 | 61.83 | **26.69** | 49.1 | 47.3 | 23.61 | 26.73 | 19.42 | 31.98 | **47.67** |
| **SegNeRF (4)** | 37.46 | 21.1 | 44.35 | 40.04 | 29.44 | 50.81 | 62.49 | 24.12 | 49.36 | 49.69 | 22.3 | 27.38 | 23.08 | 32.79 | 47.43 |

Table 2: **Fine-grained 3D semantic segmentation (3D part-category mIoU%).** Quantitative results for fine-grained 3D semantic segmentation are presented, with a detailed breakdown by category. Our method outperforms the 2D-supervised methods by significant margins in all categories, and attains comparable results with 3D-supervised methods. Methods that have access to 3D supervision (point-based methods) are highlighted in gray.

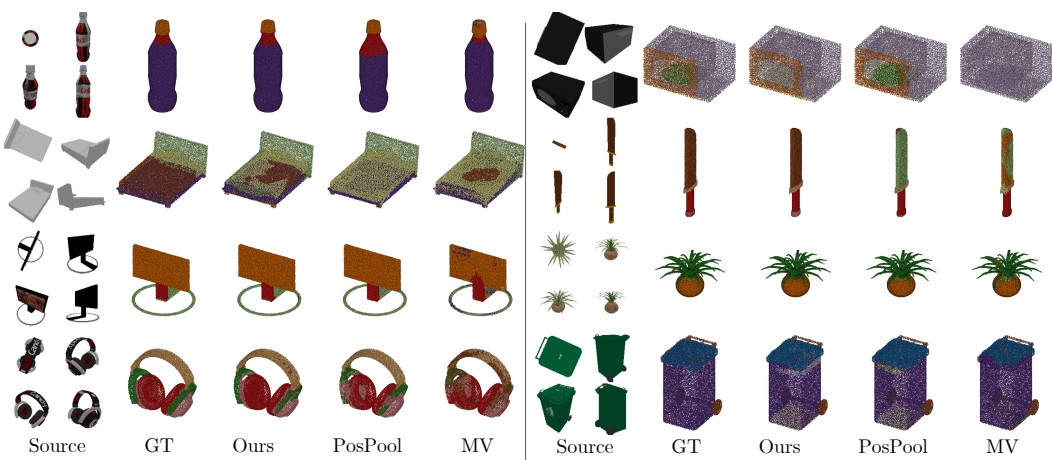

Source   GT   Ours   PosPool   MV     Source   GT   Ours   PosPool   MV

Figure 3: **3D Semantic Part Segmentation from Multiple Images.** Qualitative results for 3D semantic part segmentation from four input source views.

views, and occluded points in each view are not considered in the voting. For points occluded in all views, we take the label of the nearest neighbor point that has been seen at least in one view. MV is a more comparable baseline than point-based methods, since it is also trained *exclusively* using image supervision. Note that MV (25 images) is an upper bound baseline, since it has access to all semantic segmentation observations while our model only needs a couple of views.

**Results.** Quantitative. Table 2 reports mIoU for all methods. We find that SegNeRF performs better than all methods that are supervised *only* with images (bottom of table). In particular, SegNeRF outperforms all such baselines in *every* single class, suggesting that 3D awareness, as delivered through volume rendering, leads to improved performance. Interestingly, even when comparing MV's 25-image upper bound against SegNeRF's 1-image lower bound, we find that SegNeRF obtains better average performance. Moreover, when compared to point-based methods (bottom of the table)— and although SegNeRF lacks access to 3D supervision, we find that it performs comparatively well. Specifically, SegNeRF outperforms two point-based methods (*i.e.* PointNet and PointNet++) on average, and even outperforms the top-performing method (PosPool) on two classes. Qualitative. We show qualitative results in Figure 3. As expected, access to texture information delivers consistent improvements in segmentation performance on textured parts. This phenomenon is illustrated in the bottle and knife in Figure 3. PosPool struggles to segment the bottle's neck and the knife's sheath while SegNeRF segments these parts seamlessly. Note that PosPool and point-based methods in general have an advantage in this dataset due to the existence of points enveloped within objects, as seen with the green plate inside the microwave in Figure 3.

## 5 DISCUSSIONS

**Volume *vs*. Surface Rendering.** The task of semantic segmentation requires predicting semantic classification of points on the surface of opaque objects. It is intuitive to attempt to obtain semantic segmentation from a single point on the object surface rather than a volumetric integration along the whole ray that considers the possible existance of semi-transparent volumes. Inspired by this intuition, we experimented with replacing the volume rendering formulation shown in equation 5 with surface rendering. However, due to the sparse supervision in space given by surface rendering, we found that volume rendering leads to better performances. We defer the details and experiments on surface rendering to the Appendix.

**SegNeRF with single images in the wild.** Finally, we experiment with images taken in the wild to assess our model's generalization ability in performing both reconstruction and segmentation from a single image. Images are pre-processed to remove the background using an off-the-shelf method since we train with synthetic renderings using white backgrounds. We show the 3D reconstruction and part segmentation results in Figure 4. We observe that our model has learnt to reconstruct parts that aren't visible in the images such as the back legs. The delicate structures of the chair, such as the office chair's wheels and the garden chair's holes, are recovered. However, even if the office chair is not perfectly reconstructed, our method's segmentation ability is unaffected. Our method continues to provide accurate part segmentation.

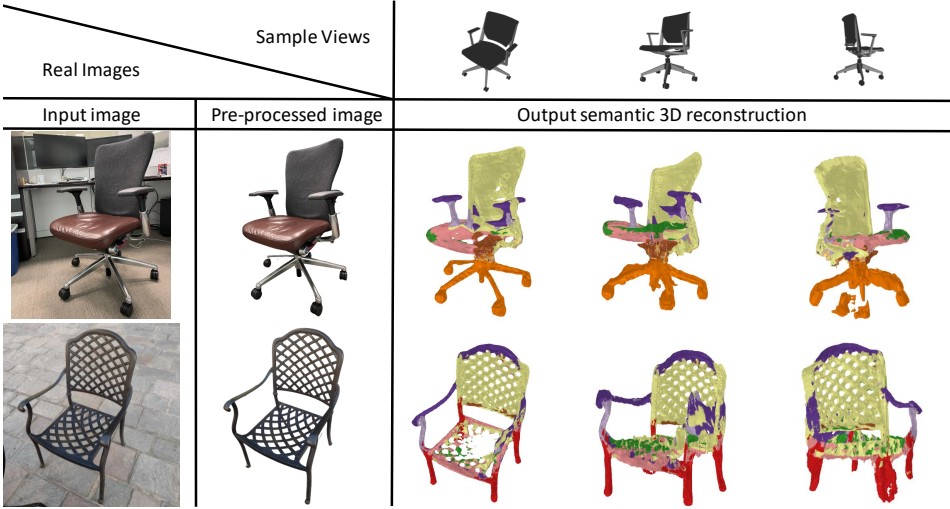

Figure 4: **3D reconstruction and part segmentation on single-view real images.** Qualitative results of two chairs given single-view images. From left to right: Original image, pre-processed image with background removed, semantic reconstruction from three different points of view: input view, front view, and back view. Notice the ability to recover and segment fine-detailed structures as well as unseen parts such as the back leg of the second chair.

## 6 CONCLUSION

In this work, we presented SegNeRF, a model for predicting joint appearance, geometry, and semantic fields for objects supervised only with images. We observe that NeRF-like representations are capable of learning more than radiance fields through volume rendering with image supervision, as shown with the semantic fields. We demonstrated the efficacy of SegNeRF on the tasks of 2D novel view segmentation and 3D part segmentation from single and multiple images. From a single image in the wild of a single object, we demonstrate a practical application of SegNeRF in performing a semantic reconstruction which could be valuable for autonomous navigation. We envision future work investigating segmentation at a larger scale (*e.g.* scene segmentation) and embedding more than just semantic information (*e.g.* instance-level segmentation).

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

## A  DATASET DETAILS

We present the abbreviation for some of the categories reported in the main manuscript in Table 3.

| Abbreviation | Bott | Dish | Disp | Ear | Fauc | Micro | Stora |
|---|---|---|---|---|---|---|---|
| Full Name | bottle | dishwasher | display | earphones | faucet | microwave | storage furniture |

Table 3: **Category abbreviation.** The complete name for the reported categories in the main manuscript.

We give additional details on the number of distinct shape instances remaining from each category remaining after matching objects from PartNet with their corresponding objects in ShapeNet in Table 4. We maintain above 90% categories such as: chair, clock, display, faucet, storage furniture, table, trash, and vase. On the other hand, these categories are below 90%: bed, bottle, dishwasher, earphones, knife, and microwave.

| | Total | Bed | Bott | Chair | Clock | Dish | Disp | Ear | Fauc | Knife | Micro | Stora | Table | Trash | Vase |
|---|---|---|---|---|---|---|---|---|---|---|---|---|---|---|---|
| **PartNet** | 22508 | 248 | 519 | 6400 | 579 | 201 | 954 | 247 | 708 | 384 | 212 | 2303 | 8309 | 340 | 1104 |
| **Ours** | 21468 | 192 | 435 | 6311 | 552 | 82 | 927 | 68 | 648 | 326 | 120 | 2243 | 8179 | 321 | 1064 |
| **Percentage** | 95.3 | 77.4 | 83.8 | 98.6 | 95.3 | 40.7 | 97.8 | 27.5 | 91.5 | 84.9 | 56.6 | 97.3 | 98.4 | 94.4 | 96.4 |

Table 4: **PartNet subset statistics.** Number of distinct shape instances for the complete PartNet dataset and the subset utilized in this paper. Additionally, the percentage of the comparison between the complete PartNet dataset and ours.

The categories that are not taken into account for the experiments are: bag, bowl, door, hat, key lamp, laptop, mug, fridge, and scissors. The total number of instances that are within these categories is 4163 (15% from the original PartNet dataset).

## B    TRAINING DETAILS

### B.1    SEGNERF ARCHITECTURE

We follow the training procedure from Yu et al. (2021b), the encoder $\mathcal{E}$ is a ResNet34 backbone and extract a feature pyramid by taking the feature maps before to the first pooling operation and after the first ResNet 3 layers. Network $\mathcal{H}$ is composed of 3 fully connected layers while network $\mathcal{G}$ is made up of 2 fully connected layers. All layers use residual connections and have a hidden dimension of $512$. We use ReLU activations before every fully connected layer. The positional encoding comprises six exponentially increasing frequencies for each input coordinate.

We train for 600000 iterations, which took roughly 6 days on 4 V100 NVIDIA GPU.

### B.2    2D NOVEL-VIEW PART SEGMENTATION

**DeepLab v3.** We train one model per category for 20 epochs (max. one day in 4 V100 NVIDIA GPUs) with a batch size of 100. Additionally, the model expects input normalized images, *i.e.* mini-batches of 3-channel RGB images of shape (N, 3, H, W), where N is the number of images, H and W are the height, and width of the images. The images have to be loaded in to a range of [0, 1] and then normalized using mean = [0.5, 0.5, 0.5] and std = [0.5, 0.5, 0.5].

DeepLab v3 returns an OrderedDict with two Tensors of the same height and width as the input Tensor, but with 21 classes. output['out'] contains the semantic masks, and output['aux'] contains the auxiliary loss values per-pixel. We remove the last layer of 21 classes and replace it with the number per category. In inference mode, output['aux'] is not useful. So, output['out'] is shaped (N, C, H, W). C being the classes per category and the desired resolution. The resolution was the finest level (3).

We utilize a ResNet-50 backbone, and the model was pre-trained on COCO 2017. Lastly, we use Adam optimizer, and cosine annealing LR scheduler with $T\_max = 10$ were used.

### B.3    SINGLE/MULTI-VIEW 3D PART SEGMENTATION

**PointNet.** and **PointNet++.** We follow the training procedure from Qi et al. (2017a) and Qi et al. (2017b), respectively. We train only one model for all categories for 251 epochs (2 days in 1 V100 NVIDIA GPU). The batch size was 45 and 25 for PointNet and PointNet++ models, respectively. Additionally, each model expects as input normalized point clouds, *i.e.* mini-batches of objects in a range of [0, 1]. We utilize Adam optimizer with learning rate 0.001, betas of $(0.9, 0.999)$ and weight decay of $0.5$. Both models were initialized with Xavier uniform initialization. Lastly, we use PointNet++ MSG version.

**PosPool.** We follow the training procedure from Liu et al. (2020), and train their architecture variant which uses sinusoidal positional embedding, PosPool*.

### B.4    SINGLE/MULTI-VIEW 3D RECONSTRUCTION

**SoftRas.** We follow the training procedure from Liu et al. (2019a), we train one model for *chair*, *table*, and all categories. In the main manuscript, the reported results are with *chair* and *table* models that were trained for 250000 iterations. Here, we detail results for *all* categories model that was trained for 500000 iterations. We utilize Adam optimizer with learning rate 0.0001, betas of $(0.9, 0.999)$.

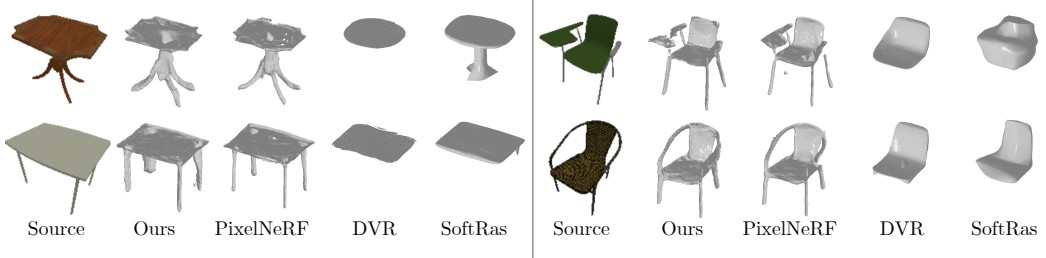

Figure 5: **3D Reconstruction from Single Image.** Qualitative results for 3D reconstruction are presented for the *chair* and *table* categories.

## C    3D RECONSTRUCTION

Obtaining a 3D reconstruction of an object from a single or few images is an ill-posed problem since there can be infinite ways in which occluded portions of an object could be structured in 3D. However, most objects tend to follow similar distributions amongst their class. For example, chairs tend to have four legs, a seat, and a backrest. Thus, neural models should be able to generate a 3D model from a single observation of an object by learning object class distributions.

In a real-world scenario, it is more likely to have access to a set of images rather than a point cloud. Thus, if one would like to use our method to get the 3D part segmentation of an object in the wild, it is necessary first to obtain the 3D reconstruction from the taken views, which can also be obtained with SegNeRF. After having the point cloud, each point can be queried in SegNeRF to obtain its segmentation prediction. Thus, this section shows an extensive 3D reconstruction task analysis in which SegNeRF could be helpful in real-world scenarios.

For the task of 3D reconstruction, we generate an object's 3D mesh given a set of posed input images of the object. To generate this mesh, we follow the standard practice in constructing a set of query points as a grid within a volume at a defined input resolution (Mildenhall et al., 2020) as detailed in Figure 1. In this work, we improve the extracted representation by leveraging the predicted semantic field, thus removing the predicted points as background by setting their density to 0. A mesh can then be extracted from this grid by applying a density threshold, and then executing marching cubes (Lorensen & Cline, 1987).

***Single-View Reconstruction.*** Here we show that our model is capable of learning such distributions for a known object class, and can generate good 3D reconstructions. We compare against Soft-Ras (Liu et al., 2019a) and DVR (Niemeyer et al., 2020a), two popular 2D-supervised single-view reconstruction methods. We follow each of their training procedures using our training data. A single model is trained for each category. Quantitative results such as chamfer distance, f-score, precision, and recall are reported in the Appendix.

Figure 5 provides evidence that PixelNeRF and our method can reconstruct high-quality geometry for complex objects, unlike existing single-view reconstruction models. Both methods can learn intrinsic geometric and appearance priors, which are reasonably effective even for parts not seen in the observation (*e.g. chair* and *table* legs). Similarly, SoftRas performs poorly in producing thin parts of the objects since it deforms a sphere mesh of a fixed size.

It also could be observed that the PixelNeRF reconstruction can provide finer details than our method. For example, the table reconstructed by PixelNeRF on the bottom-left side has a leg size more consistent with the object. Equally, some chair parts are missing in SegNeRF reconstructions. While our method does not achieve the same performance as the PixelNeRF baseline, we believe that the 3D segmentation capability comes at the price of slightly worse reconstruction.

|  | **Chair** | | | | **Table** | | | |
|---|---|---|---|---|---|---|---|---|
|  | CD↓ | F-Sc↑ | P↑ | R↑ | CD↓ | F-Sc↑ | P↑ | R↑ |
| SegNeRF (1) | 3.82 | **79.31** | 75.18 | **84.72** | 5.07 | **79.84** | 75.29 | **86.16** |
| UniSeg (1) | **2.70** | 72.4 | **75.99** | 70.23 | **3.45** | 71.26 | **87.43** | 62.41 |
| SegNeRF (2) | **1.73** | **86.53** | **84.94** | **88.72** | **1.74** | **85.96** | 83.79 | **88.92** |
| UniSeg (2) | 2.21 | 78.91 | 83.21 | 75.9 | 2.79 | 77.76 | **89.18** | 70.75 |
| SegNeRF (4) | **1.54** | **90.06** | **90.21** | **90.28** | **1.48** | **90.33** | 89.99 | **91.08** |
| UniSeg (4) | 2.38 | 79.13 | 85.68 | 74.29 | 3.09 | 77.56 | **90.87** | 69.63 |

Table 5: **3D Reconstruction Surface Rendering *vs*. Volume Rendering.** Quantitative results for 3D reconstruction are presented, with the varying number of input images represented by the number in parentheses. Volume rendering results in better reconstruction for multi-image settings, while surface rendering results in a higher precision single-image reconstruction at the cost of a lower recall. Metrics set key: **CD**: Chamfer distance., **F-Sc**: F-Score, **P**: Precision, **R**: Recall.

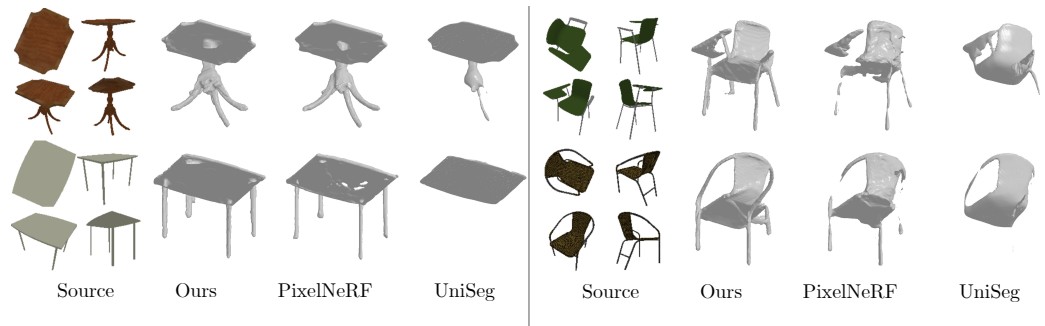

| Source | Ours | PixelNeRF | UniSeg | Source | Ours | PixelNeRF | UniSeg |

Figure 6: **3D Reconstruction from Multiple Images.** Qualitative results for 3D reconstruction from four images are provided for the *chair* and *table* categories.

## D    ADDITIONAL RESULTS

***Volume vs. Surface Rendering.*** We present a variant of our method dubbed **UniSeg**, which relies on surface rendering instead of volume rendering to generate colored and semantic images. The task of semantic segmentation requires predicting classification only on object surfaces. It is intuitive to attempt to obtain semantic segmentation from a single point on the object surface rather than an integration along the whole ray that emanates from each pixel. Inspired by (Oechsle et al., 2021), we experiment with replacing the volume rendering formulation shown in equation 5 with surface rendering. The density $\sigma$ becomes discretized into a binary occupancy representation for surface rendering. Rendering then becomes an assignment of the first surface value encountered as shown in equation 9, instead of having contributions to the final field values from all positions along a ray.

$$S(\mathbf{r}) = \sum_{i=1}^{N} s\left(\mathbf{r}(t_i)\right)\,\sigma\left(\mathbf{r}(t_i)\right)\prod_{j<i}[1 - \sigma\left(\mathbf{r}(t_i)\right)] \tag{9}$$

**Results.** Although surface rendering should intuitively benefit semantic segmentation in 3D, we observe the opposite. When using surface rendering, the average mIoU across all object categories is lowered from 32.44% to 26.92% ($-5.51\%$) for the single image setting. As shown in Table 5 and Figure 6, the reconstruction recall (R) tends to be lower when using surface rendering while the precision (P) is comparable. The lower recall and higher precision can be observed in the qualitatively thinner and smoother surfaces generated by the surface rendering method. Inspecting the Chamfer Distance (CD), we observe that the volume rendering benefits from more views, while the surface rendering saturates after two views. We believe that the full volume rendering has more learning capabilities than the surface one.

### D.1 2D NOVEL-VIEW PART SEGMENTATION

In Figure 7 and Figure 8, we show additional result for *table* and *chair* categories. These examples were randomly selected.

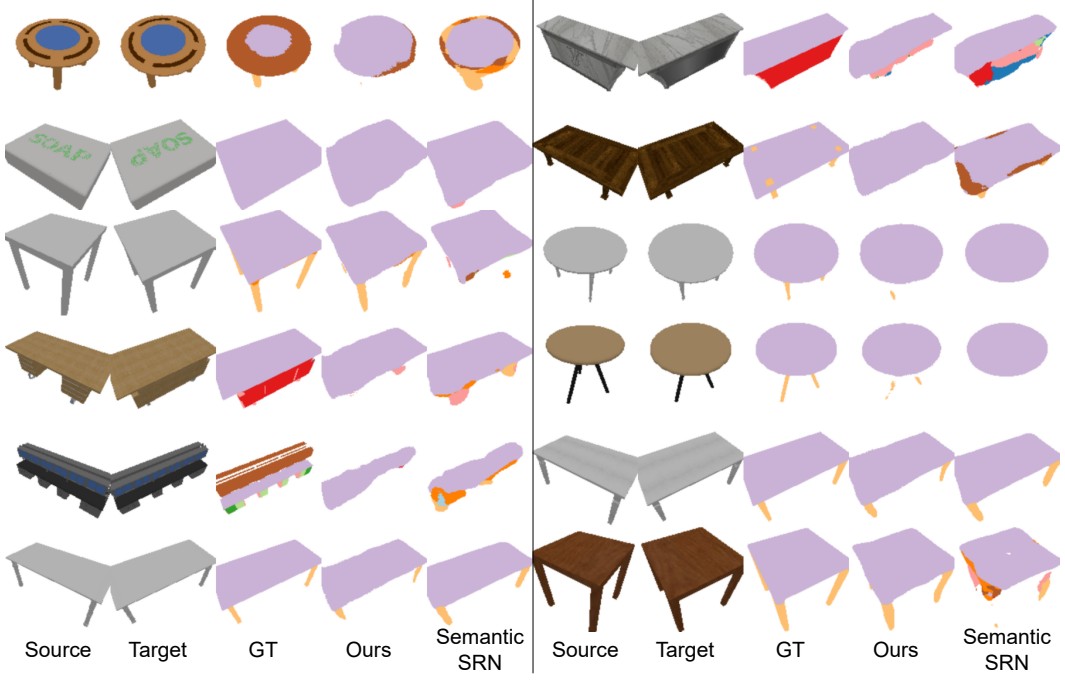

Figure 7: **Additional 2D semantic part segmentation for novel views (*table*) results.** Qualitative results for 2D semantic part segmentation for novel views.

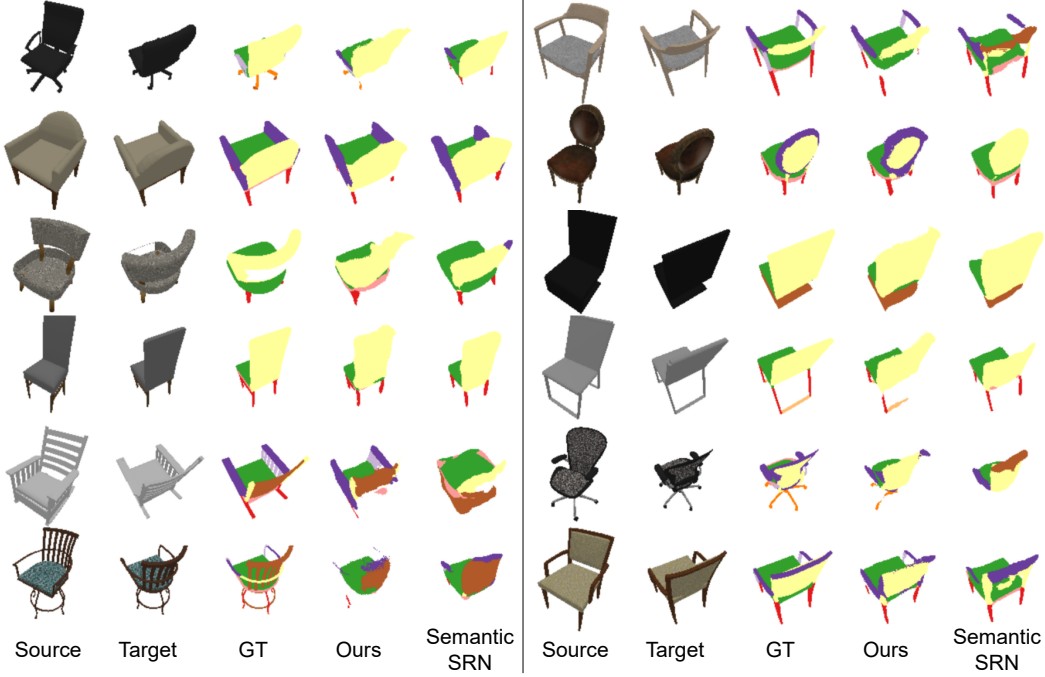

Figure 8: **Additional 2D semantic part segmentation for novel views (*chair*) results.** Qualitative results for 2D semantic part segmentation for novel views.

## D.2 Single/Multi-View 3D Part Segmentation

In Figure 9, we show additional result for all categories. These examples were randomly selected.

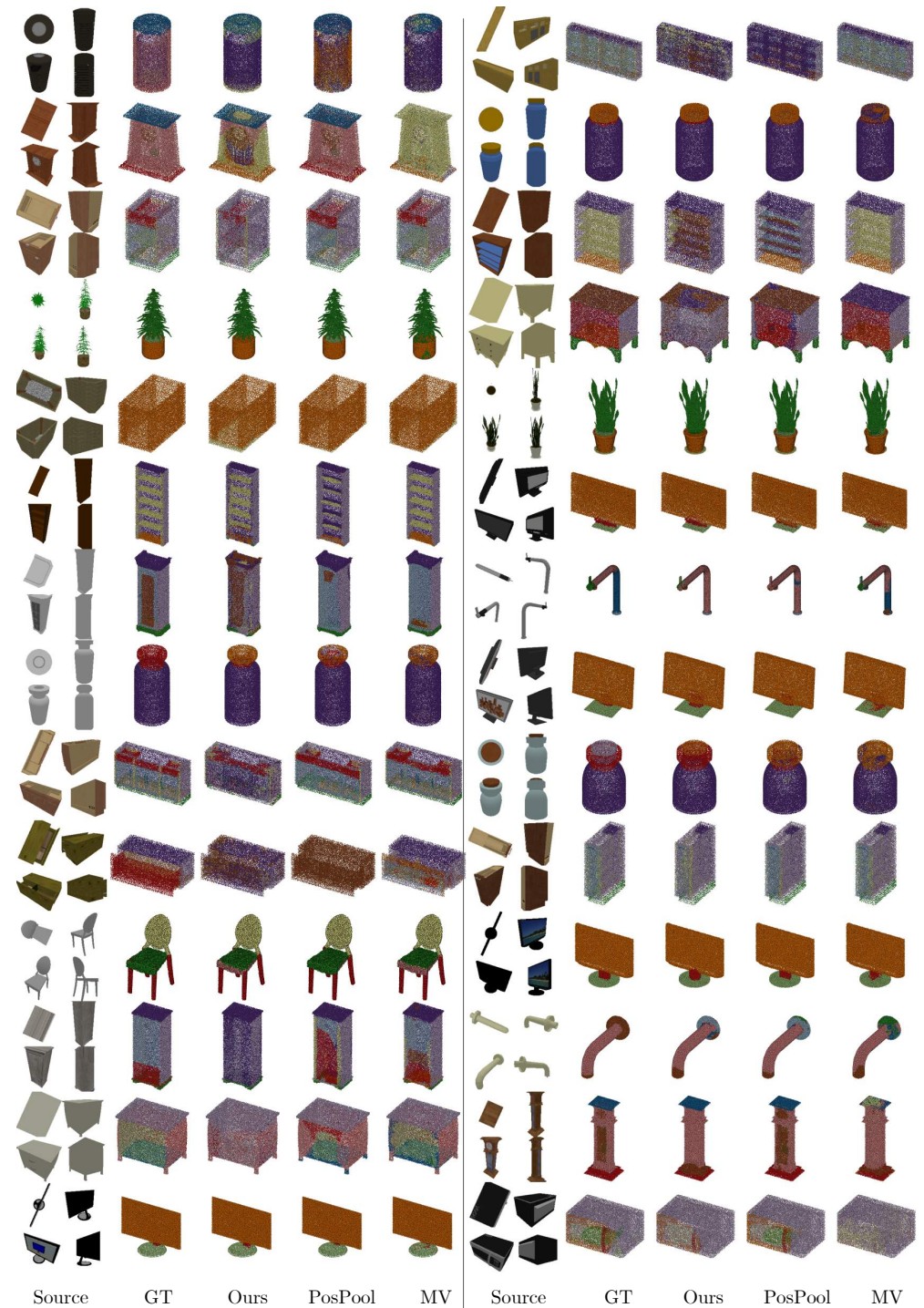

Source    GT    Ours    PosPool    MV      Source    GT    Ours    PosPool    MV

Figure 9: **Additional 3D Semantic Part Segmentation from Multiple Images Results.** Qualitative results for 3D semantic part segmentation from four input source views.

Table 6 and Table 7 report accuracy and mean accuracy for all methods.

|           | Avg   | Bed   | Bott  | Chair | Clock | Dish  | Disp  | Ear   | Fauc  | Knife | Micro | Stora | Table | Trash | Vase  |
|-----------|-------|-------|-------|-------|-------|-------|-------|-------|-------|-------|-------|-------|-------|-------|-------|
| **MV (1)**      | 61.91 | 25.71 | 90.78 | 65.58 | 51.40 | 65.17 | 88.45 | 51.92 | 63.05 | 60.61 | 76.40 | 39.50 | 57.70 | 50.33 | 80.17 |
| **MV (2)**      | 65.48 | 29.87 | 92.39 | 68.69 | 56.26 | 68.61 | 92.71 | 53.75 | 68.09 | 63.84 | 75.95 | 47.12 | 63.86 | 52.60 | 83.01 |
| **MV (4)**      | 66.06 | 33.18 | 89.72 | 71.75 | 55.55 | 67.15 | 92.97 | 55.42 | 67.50 | 66.98 | 78.87 | 46.23 | 63.49 | 52.71 | 83.35 |
| **MV (25)**     | 69.00 | 34.44 | 92.94 | **74.54** | 60.25 | 74.02 | 93.81 | 62.99 | 71.09 | 62.87 | **82.13** | 54.20 | 64.48 | 53.94 | 84.26 |
| **SegNerf (1)** | 70.70 | 33.15 | 94.01 | 70.97 | 64.10 | 83.38 | 92.50 | 64.64 | 72.23 | 74.21 | 78.74 | 58.91 | 63.64 | 54.95 | 84.38 |
| **SegNerf (2)** | 72.76 | 37.49 | 94.21 | 73.03 | 64.35 | 84.57 | 94.05 | **66.72** | 74.63 | 77.60 | 79.15 | 63.49 | 67.05 | 57.10 | 85.23 |
| **SegNerf (4)** | **73.42** | **39.58** | **94.5** | 74.37 | **64.45** | **85.6** | **94.30** | 65.18 | **74.92** | 79.49 | 77.32 | **64.47** | **68.89** | **59.32** | **85.55** |
| **PointNet**    | 76.09 | 45.35 | 89.36 | 74.70 | 71.54 | **88.23** | 96.10 | 65.67 | 72.97 | 58.05 | 89.33 | 74.75 | 73.12 | 74.66 | 91.44 |
| **PointNet++**  | 78.15 | 51.52 | **91.6** | 78.27 | 67.82 | 87.86 | **96.95** | 65.74 | 77.04 | **67.47** | 88.50 | 78.84 | **77.24** | 72.16 | **93.04** |
| **PosPool**     | 79.88 | **63.14** | 90.94 | **80.08** | **72.18** | 86.74 | 96.28 | **70.64** | **77.24** | 67.38 | **90.85** | **80.44** | 75.03 | **78.97** | 88.49 |

Table 6: **Fine-grained 3D semantic segmentation (3D part-category accuracy%).** Quantitative results for fine-grained 3D semantic segmentation are presented, with a detailed breakdown by category. Our method outperforms the 2D-supervised methods by significant margins in all categories, and attains comparable results with 3D-supervised methods.

|           | Avg   | Bed   | Bott  | Chair | Clock | Dish  | Disp  | Ear   | Fauc  | Knife | Micro | Stora | Table | Trash | Vase  |
|-----------|-------|-------|-------|-------|-------|-------|-------|-------|-------|-------|-------|-------|-------|-------|-------|-------|
| **MV (1)**      | 31.85 | 21.05 | 52.52 | 33.81 | 25.25 | 21.87 | 40.47 | 33.52 | 48.08 | 37.60 | 23.29 | 23.41 | 16.00 | 29.88 | 39.09 |
| **MV (2)**      | 37.90 | 23.53 | **53.18** | 40.81 | 25.87 | 18.39 | 58.70 | 37.31 | 50.97 | 39.95 | 21.64 | 24.57 | 38.60 | 34.36 | 62.68 |
| **MV (4)**      | 35.98 | 25.01 | 38.93 | 44.31 | 23.14 | 15.97 | 59.72 | 30.72 | 51.08 | 44.84 | 18.33 | 24.08 | **35.62** | 34.33 | **57.68** |
| **MV (25)**     | 36.35 | 28.74 | 47.07 | 45.17 | 25.62 | 19.19 | 58.95 | 43.90 | 53.97 | 41.55 | 18.36 | 30.04 | 22.39 | 32.28 | 41.66 |
| **SegNerf (1)** | 45.52 | 32.06 | 46.19 | 49.44 | 32.16 | 58.97 | 67.78 | 38.95 | 58.42 | 55.23 | **40.12** | 33.53 | 24.22 | 47.00 | 53.17 |
| **SegNerf (2)** | 49.35 | 35.76 | 47.42 | 51.98 | 34.34 | 61.85 | 73.08 | 56.28 | **63.38** | 60.29 | 37.64 | 37.08 | 29.69 | 47.64 | 54.53 |
| **SegNerf (4)** | 50.55 | 39.31 | 49.58 | 53.65 | **38.98** | 64.45 | 73.73 | **57.61** | 62.99 | 61.98 | 32.99 | **37.4** | 33.85 | 47.76 | 53.41 |
| **PointNet**    | 45.16 | 35.33 | 33.39 | 45.64 | 34.70 | 52.07 | 83.54 | 40.71 | 55.49 | 33.50 | 34.99 | 43.03 | 40.98 | 46.29 | 52.54 |
| **PointNet++**  | 50.02 | 43.47 | **34.78** | 50.61 | 35.04 | 64.25 | **89.56** | 49.45 | 56.96 | 41.92 | 34.59 | 49.11 | 44.84 | 51.05 | 54.58 |
| **PosPool**     | 58.83 | **59.76** | 30.55 | **59.98** | 51.17 | **76.78** | 73.96 | 55.51 | **63.2** | 42.22 | **67.66** | **58.64** | **54.24** | **65.86** | **64.06** |

Table 7: **Fine-grained 3D semantic segmentation (3D part-category mean accuracy%).** Quantitative results for fine-grained 3D semantic segmentation are presented, with a detailed breakdown by category. Our method outperforms the 2D-supervised methods by significant margins in all categories, and attains comparable results with 3D-supervised methods.

### D.3 SINGLE/MULTI-VIEW 3D RECONSTRUCTION

In Figure 10 and Figure 11, we show additional result for *table* and *chair* categories. These examples were randomly selected.

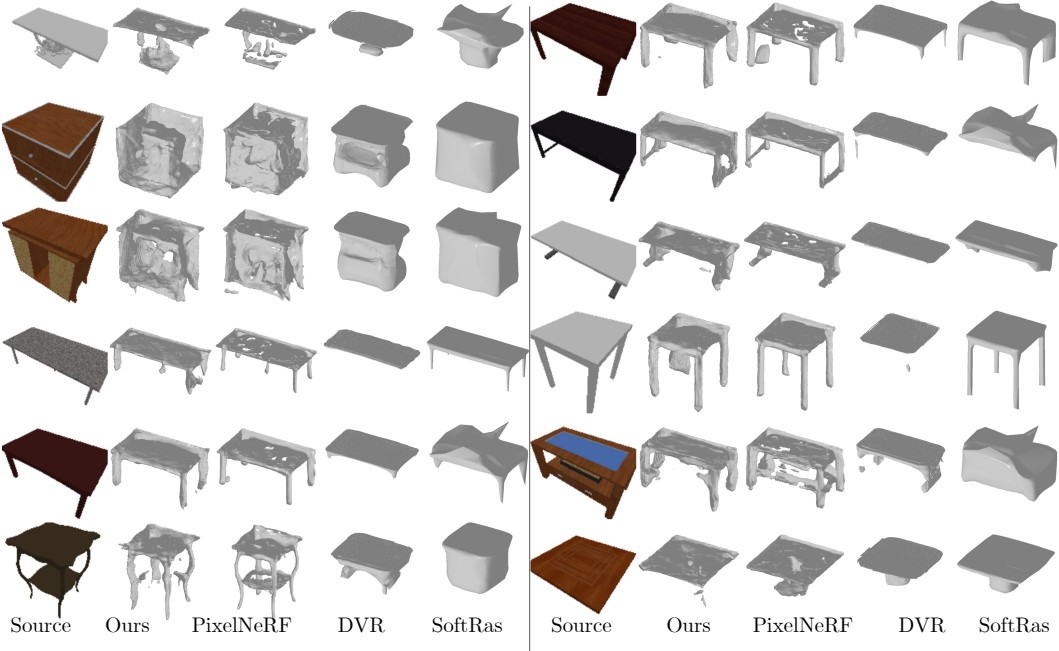

Figure 10: **Additional 3D Reconstruction from Single Image Results.** Qualitative results for 3D reconstruction are presented for *table*.

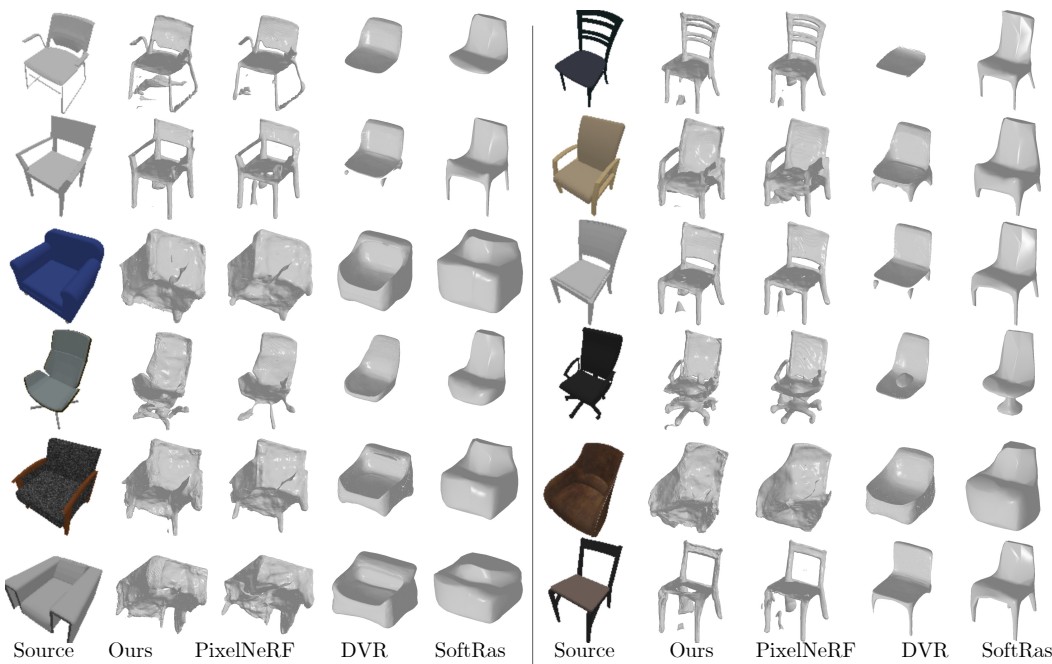

Figure 11: **Additional 3D Reconstruction from Single Image Results.** Qualitative results for 3D reconstruction are presented for *chair*.

## D.4 SEMANTIC RECONSTRUCTION

We combine 3D reconstruction and semantic segmentation by first reconstructing objects into a mesh. Vertices from the mesh are then colored according to the 3D segmentation predicted by SegNeRF at each of the vertex locations. The resulting is a semantic reconstruction mesh which can be segmented into each of the object's parts. Please see the supplementary video for visualizations on semantic reconstructions of chair objects from 4 source images, which showcases the ability of SegNeRF to perform joint 3D reconstruction and segmentation.

