# OpenReview forum: "SegNeRF: 3D Part Segmentation with Neural Radiance Fields"
_ICLR.cc/2023/Conference — Submitted to ICLR 2023_

### Official Review · Reviewer_iUC9 · 2022-10-14

**Confidence:** 5
**Correctness:** 2
**Technical Novelty And Significance:** 2
**Empirical Novelty And Significance:** 1
**Recommendation:** 3

**Clarity, Quality, Novelty And Reproducibility:**

This paper is clearly written, easy to follow and reproduce. But the novelty is rather limited.

**Strength And Weaknesses:**

Strength:
1. This paper is well written and the ideas are clearly illustrated.
2. It is nice to investigate NeRF-based model for a discriminative task.
3. It is good to see that SegNeRF achieves encouraging results on 2D and 3D segmentation tasks.

Weaknesses:
The major concern is the novelty of this work.
1. For the task of 2D and 3D segmentation, introducing a semantic field is no longer a novel design. For example, SemanticNeRF[1] and PanopticNeRF[2] employed the segmentation field for Scene Labelling and Understanding, FENeRF[3] and sofgan[4] designed the semantic field for 3D face generation and editing. The motivation and method of this paper are highly similar to these previous methods.

2. For the encoder-based frameworks, this paper combines ideas of IBRNet[5] and PointNeRF.

3. I think the novelty is to apply previous ideas to the task of 3D part segmentation, but I don’t think this can reach the bar of ICLR.

[1] In-Place Scene Labelling and Understanding with Implicit Scene Representation.
[2] Panoptic Neural Fields: A Semantic Object-Aware Neural Scene Representation.
[3] FENeRF: Face Editing in Neural Radiance Fields.
[4] Sofgan: A portrait image generator with dynamic styling
[5] IBRNet: Learning Multi-View Image-Based Rendering


**Summary Of The Paper:**

This paper proposes to employ neural radiance fields to model the 3D part information of objects. In parallel with the color and density information, the NeRF model also predicts the semantic segmentation probability of each point in the 3D space. This yields a semantic field that enables the model to achieve good 2D and 3D segmentation results.

**Summary Of The Review:**

The strength and weakness of this paper are clear. I don’t think this can reach the bar of ICLR because of the novelty of the idea and method.

---

> ### Author Response · Authors · 2022-11-19
> **Response**
>
> We would like to thank the reviewer for their time evaluating our work.
> 1) While previous works have used semantic fields as part of their method, this does not mean they have the same motivation and methodology as ours.
>    1) Semantic NeRF works with access to semantic labels and performs scene optimization, making it useful for offline tasks such as label denoising, while we require only one RGB view of an object for predicting 3D segmentation, making our method useful for time-constrained scenarios. Moreover, Semantic NeRF does not provide results on 3D semantic segmentation.
>    2) PanopticNeRF is part of a multi-stage pipeline in which one requires first object detection before tracking each individual object to train a separate NeRF for each. Once again, this requires expensive offline optimization and access to a track of objects, and does not enable straightforward extraction of a 3D segmentation. Our method only requires access to a single RGB image of an unseen object to perform part segmentation.
>    3) FENeRF and sofgan, as you mention, have completely different motivations, methodology, and applications for face image editing, and do not enable access to any explicit 3D segmentation.
> 2) Our encoder-based framework is mostly based on PixelNeRF, as we state clearly throughout our work.
> 3) The main novelty is showing for the first time how 3D semantic segmentation can be extracted from a segmentation field predicted from a single image of an object, without the need for any test-time optimization. This allows our method to be applicable for time-constrained segmentation applications, while only requiring access to images rather than more complex 3D scan data.

---

> > ### Comment · Reviewer_iUC9 · 2022-12-01
> > **Response to the authors**
> >
> > Thanks for the clarification. But the slight differences in motivation do not mean there is a significant contribution. I would still keep my rating.

---

### Official Review · Reviewer_zVwm · 2022-10-21

**Confidence:** 4
**Correctness:** 3
**Technical Novelty And Significance:** 2
**Empirical Novelty And Significance:** 2
**Recommendation:** 5

**Clarity, Quality, Novelty And Reproducibility:**

Mostly clear, writing of high quality except for a couple of places in the main text. Reproducibility must be easy but was not checked, as the method is a rather straightforward extension of PixelNeRF.

**Strength And Weaknesses:**

The paper is well written, the delivery is good, however, there are a few questions for the authors.

- Fig.1 is nice but does not serve the purpose of explaining the method. Color coding segmentation/reconstruction/violet objectives are counter-intuitive. The 6 H/G MLPs are not placed into the context. The chair is presented from 3 different perspectives, yet only two are present on the left side. Without the introduction of the notation, the E symbol reads as taking expectation, and not the feature map explained deep in the Method section. I would recommend a complete re-design of the figure.

- Sec. 2.1. states that voxelized representations have a cubic complexity in dimensions; this is not true in the light of works like TensoRF and other decomposition methods.

- Sec. 3.2. is titled "Predicting Volume Rendering" - what does this mean exactly? Perhaps some rephrasing is in order.

- Starting the "Next, we elaborate" paragraph in Sec. 3.2 and until the end of that section, the writing is rather negligent with notation. I would recommend checking how a similar concept was introduced in PixelNeRF and using a similar notation, more space to explain this rather critical part of the method and introduce notation before or around the first usage. In the end, I was not quite sure if the method (its part explained in the section) has any important differences from PixelNeRF, apart from the added semantic classes prediction. If not, then using PixelNeRF notation is advised.

- Related to the previous comment, PixelNeRF applies positional encoding to 3D point x, whereas in your paper $\gamma$ seems to be applied to the projected coordinates. Why was this change introduced, what was the motivation? Have you tried keeping this as in PixelNeRF?

- In Sec. 3.3 you present the function $s(I, r, d)$, meaning that semantic information at each point depends on the viewing direction $d$. How so? Shouldn't it be independent of viewing direction like e.g. optical density $\sigma$?

- Why is there a zero in Eq. (8)? Is it by any chance there to fix the issue from the previous comment?

- I'm not sure why Sec.3.4 is needed, if the function $s$ functions properly (not requiring Eq. 8 hack).

- Why is there only 81% of PartNet dataset valid for dataset creation purposes?

- View ID 135 must be clarified or moved to the supplmentary





**Summary Of The Paper:**

The paper proposes an extension of PixelNeRF to extend the neural field with auxiliary dimensions, corresponding to semantic classes of the scene. The authors propose to learn semantic fields using 2D semantic supervision and demonstrate that their approach reaches the performance of several prior approaches enjoying 3D semantic supervision (except one SOTA method).

**Summary Of The Review:**

The paper provides an important study of leveraging 2D semantic maps for learning 3D-aware semantic representations for semantic novel view synthesis using NeRF. The method seems like a small extension on top of PixelNeRF, but there are a few points to be clarified (see Weaknesses). The empirical study looks good, but may not present too much insight.

---

> ### Author Response · Authors · 2022-11-19
> **Response**
>
>
> Thank you for the thorough review.
> 1) Thank you for your comments on the figure. We will take them into account for a clearer redesign of Figure 1.
> 2) We would argue that decomposition methods are a way of alleviating issues with voxel-based representations through approximations. However, our statement still holds for explicit voxel-based representations.
> 3) With “Predicting Volume Rendering” we aim to convey the idea that we are not just memorizing the semantics of a single scene through volume rendering, but rather predicting the semantic field for novel objects while performing volume rendering. Perhaps “Predictive Volume Rendering” would be more appropriate.
> 4) Our method shares most of its formulation with PixelNeRF except for the added semantic field. However, we found the notation used in PixelNeRF to be unclear since it deviates from the notation used in other NeRF papers. We borrowed notation from PixelNeRF where we saw fit, but explained our method in a manner in line with NeRF papers.
> 5) As an example of the unclear formulation of PixelNeRF we may borrow your following comment. We perform positional encoding in the same manner as PixelNeRF: on the projected coordinates of each camera’s coordinate system. This fact gets lost in PixelNeRF’s methodology, but becomes clear with our notation.
> 6) It is correct that 3D segmentation does not depend on the viewing direction, which is why we fix the input direction to 0 as shown in Eq. 8. However, we decided to keep the general formulation including viewing direction to keep a consistent notation with the related radiance field “c”.
> 7) Section 3.4 details how the semantic field which was learnt using volume rendering with 2D image supervision can be later reused for the different task of 3D semantic segmentation by querying it at independent points rather than along rays. We would like to reiterate that this is different from novel view synthesis, since we extract segmentation at single points in 3D space.
> 8) As detailed in section 4, we require having matching textured models from ShapeNet in order to have both RGB images extracted from ShapeNet along with semantic images from PartNet. Only 81% of the objects in PartNet have a corresponding textured model in ShapeNet.
> 9) We explain that the input front-facing image corresponds to view ID 135 in our dataset of rendered images. However, we will move this detail to the supplementary material.

---

### Official Review · Reviewer_miT3 · 2022-10-24

**Confidence:** 4
**Correctness:** 3
**Technical Novelty And Significance:** 2
**Empirical Novelty And Significance:** 2
**Recommendation:** 5

**Clarity, Quality, Novelty And Reproducibility:**

Clarity: 8/10
Quality: 7/10
Novelty: 6/10
Reproducibility: 9/10 if the code released

Some experimental details are not clear:
1. For the PartNet dataset, do you use all three levels (e.g., from coarse to fine-grained). According to the visual comparison, only coarse-grained level parts are used?
2. The authors say that they re-run all baseline methods. However, In Table 1, Semantic SRN is only evaluated in two categories. Is there any explanation for that?
3. "Once DeepLab v3 is trained, we evaluate it on the validation set generated by PixelNeRF consisting of 25 novel poses, and rendered from a single view of the object instance using a category-specific
PixelNeRF." What does render mean here? It's kind of confusing.
4. "Furthermore, Figure 2 also illustrates how PixelLab mislabels fine structures." However, PixelLab is not shown in Figure 2.
5. The training details of 3D point-based methods are missing? Training/test split, input data format (xyz, or xyz + rgb, #points), etc.
6. The multi-view baseline in section 4.2 is very confusing. How is the DeepLabV3 trained? Where does the input image come from during inference? How is the voting actually performed?
7. Another alternative multi-view baseline is directly rendering the input point cloud and then using 2D segmentation networks on the rendered 2D images. In this way, we can render an arbitrary number of 2D images instead of being limited to 25 images.


**Strength And Weaknesses:**

Strengths:
1. The idea of training an image-conditioned NeRF together with a semantic field is simple and straightforward.
2. The paper is well-written and easy to follow.

Weaknesses:
1. The idea of training a semantic field in NeRF is not new. Many prior works (e.g., Semantic NeRF, NeSF, and Semantic SRN) share a similar idea, which limits the novelty of the proposed method.
2. I think NeSF is very similar to the proposed method and fail to find any major difference between the two methods. However, NeSF is missing in the experimental comparison.
3. In equation (5), why does the semantic field function s relies on the view direction d? From my understanding, the semantic label of a 3D point should not change with regard to the view direction.
4. The current method is not tested within each object category. It would also be interesting to see whether the model trained on a shape category can generalize to similar categories as well, since we cannot train separate models for all shape categories in our real-world applications.
5. Some experimental details are unclear or missing.

**Summary Of The Paper:**

This paper proposes a method for 3D shape part segmentation. It learns an image-conditioned NeRF together with a semantic field. The method enables novel view (2D image) part segmentation and 3D point cloud part segmentation. The authors evaluate the proposed method on the PartNet dataset and show some real-world demos.

**Summary Of The Review:**

I fail to tell much difference between the proposed method and existing methods (e.g., NeSF). I strongly suggest the authors provide a more thorough discussion and comparison with the existing semantic NeRF works. Otherwise, I would prefer to reject the submission.

---

> ### Author Response · Authors · 2022-11-19
> **Response**
>
> Thanks for the insightful review.
> 1) While a handful of prior works have investigated semantic segmentation with neural fields, there are significant differences which make this work novel. As opposed to Semantic NeRF and Semantic SRN, we evaluate 3D semantic segmentation and not just novel view segmentation. Additionally, we focus on predicting 3D semantic segmentation of novel scenes from single/few views without requiring optimization during inference, while both Semantic NeRF and Semantic SRN require optimization at inference time to extract segmentation.
>
> 2) Our work has several major differences with respect to NeSF.
>    1) We tackle a different task: single/few-view 3D segmentation. While NeSF takes a pre-trained NeRF model as input during inference, which requires 210 RGB images to train, we require as few as a single image to perform inference.
>    2) Inference efficiency. At inference time for novel scenes, we require no optimization - simply a forward pass through a convnet along with a forward pass through the semantic field MLP. In contrast, NeSF requires optimizing a NeRF model for 25k iterations prior to performing inference.
>    3) Network architecture. NeSF leverages a 3D sparse convnet on a grid sampling of the density field from a pre-trained NeRF model, depending solely on geometry, while we focus on inferring segmentation from a 2D convnet, focusing on image appearance rather than geometry.
>
> 3) It is correct that 3D segmentation does not depend on the viewing direction, which is why we fix the input direction to 0 as shown in Eq. 8. However, we decided to keep the general formulation including viewing direction to keep a consistent notation.
>
> 4) While it would be great to have a single model for novel classes, this is still an active line of research since the task of semantic segmentation currently requires defining the semantic classes in advance. Our work does not focus on tackling this challenging task of performing semantic segmentation without advance knowledge of the objects nor semantic classes. However, in controlled scenarios in real world applications it is generally possible to define a set of objects with which an agent is expected to interact and which require fine-grained interactions.
>
> 5)
>    1) As we state in the table captions, we report results using the fine-grained segmentation labels (level 3 in PartNet).
>    2) Semantic-SRN only claim their method performs well on these categories. For fairness, we only re-run their method on these same categories.
>    3) Rendering is the process of generating an image from a 3D model via an algorithm. In this context, we use render to mean that we generate images from novel views from PixelNeRF, which has an implicit 3D model.
>    4) Thank you for pointing out this error. The figure was edited in a prior revision of the paper due to size constraints, but this will be corrected in the next revision.
>    5) We utilize the same object train/test splits for all methods. These are derived from the PartNet dataset with the exception of the removed objects due to missing ShapeNet models. For point-based methods we used xyz + normals as input, which tends to perform best. The same point clouds provided for validation from PartNet are used for both point-based networks and our method, which contain 10,000 points. We will add these details to the supplementary material.
>    6) We use the same DeepLabV3 model used for the 2D novel view segmentation baseline, detailed in Section 4.1. We will clarify this in the main text. Input images are the RGB renderings of objects from the validation set, using the same views as for our method. While voting can be performed in many different ways on logits or final predictions, we found mode voting on the final predictions to perform best and use it as the baseline.
>    7) That is indeed another baseline one could try for multi-view segmentation. However, we found the number of images used to not be the limiting factor for multi-view 3D segmentation. Performance saturates with more than 25 views, so increasing the number of views up to the maximum 250 views we rendered does not improve the baseline performance.

---

> > ### Comment · Reviewer_miT3 · 2022-12-02
> > **Response to authors**
> >
> > Thanks for the clarification. I would prefer to keep my rating.

---

### Official Review · Reviewer_ynF6 · 2022-11-02

**Confidence:** 4
**Correctness:** 4
**Technical Novelty And Significance:** 2
**Empirical Novelty And Significance:** 2
**Recommendation:** 5

**Clarity, Quality, Novelty And Reproducibility:**

The idea of the work is clear and solid. The authors will release the code and data for better reproducibility. However, the main concern is that the proposed idea is not novel enough.

**Strength And Weaknesses:**

** Strengths **

1. The proposed method/representation is quite concise and technically sound. Simply integrating a new semantic field into a NeRF framework can improve the modeling capability. It shows more possibility to extend from geometry and appearence to more information.
2. They conduct extensive comparison studies to demonstrate the superior of the proposed approach in terms of semantic segmentation.
3. Code and data will be shared and they would be a nice bonus to the literature.
4. The paper is generally well written and easy to follow.

** Weaknesses **

1. The main weakness of the submission is the limited contribution it offers. The idea that integrating a semantic field into a traditional NeRF representation is straightforward and not novel enough. It looks more like an incremental work to NeRF and nothing special is proposed to address particular problems.
2. Although the main focus of the paper is semantic segmentation, we would still like to know how it performs on geometry and appearance quantitatively. If possible, I would also like to see an ablation study to show whether adding a semantic field has any positive or negative effect on the original NeRF's geometry and appearance modeling performance.
3. In Tab2, why using more views make the final reuslts worse for “Ear”, “Micro” and “Vase”? Could you provide a more detailed analysis of this observation?
4. In Fig4, could you show more novel view synthesis results instead of only semantic segmentation? Why the top region of the chair back is recognized as purple?


**Summary Of The Paper:**

The paper proposes a generalizable neural field representation that combines the traditional neural radiance field with a semantic field, which enables a simultaneous modeling the geometry, appearance and semantic information from a few images. The proposed representation enables 2D novel view segmentation and 3D part segmentation from single and multiple images. They also demonstrate the capability of 3D semantic reconstruction from single in-the-wild image.

**Summary Of The Review:**

Given the above-mentioned weaknesses (limited contribution, missing analysis, etc), my suggestion is that the paper is currently marginally below the acceptance threshold for a publication.

---

> ### Author Response · Authors · 2022-11-19
> **Response**
>
> We appreciate the thoughtful review.
> We would like to address the weaknesses as follows:
>
> 1) The main concern is that the proposed idea is not novel enough since it is understood as "integrating a semantic field into a traditional NeRF representation". However, we would like to address two main contributions which separate us from traditional NeRFs:
>     1) The ability to $\mathbf{predict}$ a semantic field for a scene never seen during training, which is required for discriminative tasks as opposed to the traditional setting of NeRF in which a single scene is represented.
>     2) Demonstrating that it is possible to use a 3D semantic field - learnt with image supervision via volume rendering - for the task of 3D semantic segmentation. We are the first NeRF method providing 3D semantic segmentation results along with a comparison against typical 3D segmentation methods.
>
> 2) In general, we found that the semantic field did not have a major effect on reconstruction of neither geometry nor appearance. Since previous works have already tackled appearance and geometry reconstruction and we didn't find significant results, we chose to leave them out of the main paper.
>
> 3) Since our method predicts segmentation for novel objects based on a learnt image encoder, there can be disagreements between feature predictions from different views - especially for objects without enough training data. The "Earphones" and "Microwave" classes are only composed of 68 and 120 object instances as shown in the appendix. Additionally, segmentation performance tends to saturate after a number of views (typically more than 4 views). For the “Vase” category, saturation occurs with only 2 views.
>
> 4) The purple category is “armrest” and, due to the appearance similarity between armrests and thick top supports, the model predicts the top bar as an armrest. Predictions depend mostly on appearance similarity rather than geometry or absolute position since the encoder is shared between rotated views of the object.

---

> > ### Comment · Reviewer_ynF6 · 2022-11-30
> > **Response to the authors**
> >
> > Thank you for the detailed explanation and clarification. However, I would still keep my score due to the limited contribution that this work offers, as also noted by other reviewers.

---

### Decision · Program_Chairs · 2023-01-20

**Decision:**

Reject

**Justification For Why Not Higher Score:**

N/A

**Justification For Why Not Lower Score:**

N/A

**Metareview: Summary, Strengths And Weaknesses:**

This paper proposed to perform 3D part segmentation using a neural field representation named SegNeRF. While the reviewers generally find the paper technically sound and the experiments are detailed. there are limited contributions on introducing a semantic field with NeRF compared to previous works. All reviewers recommended rejecting the paper and the AC agreed with the reviewers.